# Beyond Equilibrium: Non-Equilibrium Foundations Should Underpin Generative Processes in Complex Dynamical Systems

## Abstract

This position paper argues that next-generation non-equilibrium-inspired generative models will provide the essential foundation for better modeling real-world complex dynamical systems. While many classical generative algorithms draw inspiration from equilibrium physics, they are fundamentally limited in representing systems with transient, irreversible, or far-from-equilibrium behavior. We show that non-equilibrium frameworks naturally capture non-equilibrium processes and evolving distributions. Through empirical experiments on a dynamic Printz potential system, we demonstrate that non-equilibrium generative models better track temporal evolution and adapt to non-stationary landscapes. We further highlight future directions such as integrating non-equilibrium principles with generative AI to simulate rare events, inferring underlying mechanisms, and representing multi-scale dynamics across scientific domains. Our position is that embracing non-equilibrium physics is not merely beneficial—but necessary—for generative AI to serve as a scientific modeling tool, offering new capabilities for simulating, understanding, and controlling complex systems.

## 1  Introduction

Since Boltzmann introduced statistical mechanics in the 1870s, a groundbreaking framework in mathematical physics, the core philosophy of this theory has remained consistent: uncovering the statistical laws governing the motion of microscopic particles, which initially appear to move in an irregular manner. In the 1950s, Jaynes [1] extended this theory to information theory and statistical learning, proposing that if the motion of microscopic particles is viewed as data, this data must adhere to similar mathematical principles as statistical mechanics, from Boltzmann distribution [2, 3] to Energy-based models (EBMs) [4–11], from variational free energy theory [3, 12] to Variational Encoders (VAEs) [13–17], from Schrödinger bridges [18–20] to the time-reversal duality in the Diffusion model [21–23]. In the 1980s, a series of groundbreaking discoveries [4–7, 24] in artificial intelligence emerged, many of which were inspired by the principles of statistical mechanics. After a period of relative dormancy, the intersection of artificial intelligence and statistical physics has recently sparked renewed interest and innovation [20–23, 25–30].

Although generative models in machine learning are typically designed for practical tasks such as image, text, and audio synthesis, they share a core philosophical foundation with statistical physics: both seek to model and generate probability distributions from high-dimensional, noisy data. However, many real-world scientific and physical systems are inherently dynamical, exhibiting time-dependent, path-dependent, and often irreversible behavior. Capturing such complex temporal evolution remains a fundamental challenge in generative modeling, as most existing frameworks assume static data distributions or stationarity. This gap motivates a deeper integration between

Submitted to 39th Conference on Neural Information Processing Systems (NeurIPS 2025). Do not distribute.

generative modeling and statistical physics, especially in the context of dynamical systems that operate far from equilibrium. In this position paper, we argue that generative models grounded in non-equilibrium statistical mechanics provide a principled and expressive framework to simulate, predict, and understand the temporal evolution of such systems, going beyond the limitations of equilibrium-based models.

In Section 2, we introduce the connection between statistical physics and generative modeling from three aspects: equilibrium-inspired models, non-equilibrium-inspired models, and non-physics-inspired models. In Section 2.1, we show how equilibrium statistical mechanics inspires machine learning. Equilibrium systems follow Boltzmann distributions, linking state probabilities to energy. Starting with the Ising model, we introduce energy-based models (EBMs) such as Hopfield networks [4] and RBMs [5–7], which use energy functions to model data. Modern EBMs scale this approach to high-dimensional data using deep networks and Langevin dynamics [9–11]. Despite their power, EBMs face challenges including intractable normalization, misalignment with non-equilibrium data, and poor modeling of dynamics. Section 2.2 outlines how non-equilibrium physics reframes generative modeling as distribution transformation via Markov chains [19, 31, 32]. Diffusion models [25, 23] implement this through stochastic dynamics. Advances such as Fokker-Planck constraints [26], Poisson flows [27], and Schrödinger bridges [20, 28] improve sampling efficiency. In Section 2.3, we relate flow-based models [33–35], VAEs [13, 14, 16], autoregressive models [36–38], and GANs [39] to statistical physics, noting their limitations in capturing system dynamics.

In Section 3, we empirically demonstrate that non-equilibrium generative models are better suited for such systems, as they capture time-asymmetric dynamics and transient steady states more effectively. Using a modified Printz potential system with time-varying energy fields governed by overdamped Langevin dynamics, we compare two generative strategies: (i) equilibrium-based Boltzmann sampling and (ii) non-equilibrium sampling via denoising Langevin dynamics. Experiments show that the non-equilibrium approach consistently achieves lower Jensen-Shannon divergence when generating time-dependent particle distributions, better reflecting the evolving energy landscape. Unlike equilibrium methods, which rely on static distributions, non-equilibrium models adaptively track gradient flows, enabling more accurate simulation of dynamic transitions. These results underscore the necessity of non-equilibrium frameworks for modeling the temporal evolution of open, evolving systems.

In Section 4, we further discuss the combination of non-equilibrium statistical mechanics and generative AI from two aspects: (1) non-equilibrium-inspired AI models for physical science and complex dynamical systems; and (2) non-equilibrium physics for more advanced generative models.

**This position paper argues that non-equilibrium physics-inspired generative algorithms represent a new frontier for modeling dynamical systems in AI for science. Dynamical systems, ranging from molecular reactions and fluid turbulence to biological networks and climate dynamics, are inherently governed by time-asymmetric, dissipative, and stochastic processes far from equilibrium. Non-equilibrium-inspired algorithms treat data as outcomes of underlying time-evolving processes, offering a principled framework to learn, simulate, and control complex system trajectories. By integrating physical priors such as entropy production, irreversible flows, and fluctuation theorems, these models have potential to more faithfully model the real-world dynamics across scientific domains.**

## 2 Revisiting Generative Models Through Statistical Mechanics

### 2.1 Energy-based generative models inspired by equilibrium statistical mechanics

Equilibrium describes a macroscopically stable state, where the Boltzmann distribution [40] relates state probability to energy:

$$P_i = \frac{1}{Z} e^{-\frac{E_i}{kT}},$$

(1)

with $E_i$ as the energy of state $i$, $T$ the temperature, and $Z$ the partition function. This allows energy-based modeling of otherwise intractable distributions.

Building on this, energy-based generative models [4, 41, 10] approximate data distributions via energy functions. They root in the Ising model [42], where discrete spin states interact with neighbors:

$$E(\mathbf{s}) = -\sum_{\langle ij \rangle} J_{ij} s_i s_j,$$

(2)

85 with $J_{ij} > 0$ lowering energy when spins align.

86 Hopfield networks [4] generalize this to neural systems, encoding memory through Hebbian learning
87 with energy

$$E(\mathbf{s}) = -\frac{1}{2} \sum_{i \neq j} W_{ij} s_i s_j - \sum_i \theta_i s_i. \tag{3}$$

88 RBMs [41] further extend this to bipartite graphs with visible $\mathbf{v}$ and hidden $\mathbf{h}$ layers:

$$E(\mathbf{v}, \mathbf{h}) = -\sum_i b_i v_i - \sum_j c_j h_j - \sum_{i,j} W_{ij} v_i h_j, \tag{4}$$

89 learning data distributions via contrastive divergence [8]. Together, these models exemplify how
90 equilibrium statistical mechanics informs generative learning.

91 Modern energy-based models combine equilibrium statistical mechanics with deep networks and
92 Langevin dynamics to scale to high-dimensional, multi-modal data with flexible composition and
93 sampling [9–11, 43, 44]. However, equilibrium-inspired EBMs face three major limitations. First,
94 the partition function $Z$ is intractable, requiring approximations like contrastive divergence [8] or
95 MCMC [45–47], which can yield unstable training. Second, their reliance on stationary Boltzmann
96 assumptions mismatches the non-equilibrium, driven nature of real-world processes. Third, they lack
97 inherent mechanisms for modeling temporal transitions, limiting applicability to dynamic systems.

## 2.2 Generative models inspired by non-equilibrium statistical mechanics

99 While in theory any target distribution $P(\mathbf{x}) = \frac{\phi(\mathbf{x})}{Z}$ can
100 be modeled via a flexible neural network $\phi$, in practice,
101 computing the partition function $Z$ and sampling from
102 $P(\mathbf{x})$ are computationally costly. Non-equilibrium statis-
103 tical physics [19, 52, 53, 31, 54, 32, 55, 56] addresses this
104 by constructing a Markov chain that transforms a simple
105 distribution (e.g., Gaussian) into the target via diffusion.
106 This shifts the learning objective from fitting $P(\mathbf{x})$ directly
107 to modeling diffusion dynamics. By breaking the problem
108 into small local perturbations, this paradigm [21] offers a
109 more tractable alternative to directly specifying a global
110 potential.

Table 1: Generative models inspired by non-equilibrium statistical mechanics.

| Algorithms | Drift-term | Diffuse-term |
|---|---|---|
| DDPM [22] | | ✓ |
| SMLD [25] | | ✓ |
| VDM [16] | | ✓ |
| SDEs [23] | ✓ | ✓ |
| RDM [48] | ✓ | ✓ |
| Flow++ [49] | ✓ | |
| PFGM [27] | ✓ | |
| DiffFlow [50] | ✓ | ✓ |
| DSB [20] | ✓ | ✓ |
| DSBM [28] | ✓ | ✓ |
| LightSB-M [51] | ✓ | ✓ |

111 Inspired by non-equilibrium statistical physics, diffusion models systematically and gradually destroy
112 the data distribution structure through an iterative forward diffusion process. The model then
113 learns the reverse diffusion process to recover the data structure [22, 25]. Once the score function
114 $s_\theta(\mathbf{x}) = \frac{\partial \log P(\mathbf{x})}{\partial \mathbf{x}}$ is trained, Langevin dynamics $\mathbf{x_{i+1}} = \mathbf{x_i} + \epsilon \frac{\partial \log P(\mathbf{x})}{\partial \mathbf{x}} + \sqrt{2\epsilon} \mathbf{z_i}$ [24, 57] are
115 iteratively applied to sample from the distribution, where $\mathbf{z_i} \sim \mathcal{N}(0, I)$. Song et al. [23] further
116 unified the continuous-time evolution between the data distribution and the prior Gaussian distribution
117 into a set of forward and backward stochastic differential equations (SDEs) (see Appendix A)

$$\begin{cases} \mathrm{d}\mathbf{x_t} = f(\mathbf{x_t}, t)\mathrm{d}t + \sigma(t)\mathrm{d}W_t, \\ \mathrm{d}\mathbf{x_t} = \left[ f(\mathbf{x_t}, t) - \sigma^2(t) \frac{\partial \log P(\mathbf{x}, t)}{\partial \mathbf{x}} \right] \mathrm{d}t + \sigma(t)\mathrm{d}\tilde{W}_t, \end{cases} \tag{5}$$

118 where the choice of drift coefficient $f(\mathbf{x}, t)$ defines different SDEs, and $W_t$ is the standard Wiener
119 process scaled by the diffusion coefficient $\sigma(t)$.

120 The unified differential equation form of diffusion models has inspired further integration of non-
121 equilibrium statistical physics. From the perspective of stochastic processes, the forward process
122 described in Equation 5, which represents the spatiotemporal evolution of data density, can be
123 characterized by the classical Fokker-Planck equation (FPE) [58]. Therefore, in principle, the data
124 density at all time steps can be recovered by solving the FPE, requiring only guidance of the noise
125 density without additional learning [59]. However, in the work of Lai et al. [26], they found that
126 the score function optimized through score matching [60, 61] fails to satisfy the density distribution
127 derived from the FPE spontaneously. By incorporating a regularization term loss guided by the FPE
128 as a physical prior, diffusion models can provide more accurate density estimates for synthetic data.

To reduce the computational burden of classical diffusion processes, recent work explores more efficient diffusion paths. Xu et al.[27] proposed Poisson flow generative models, initializing from a uniform distribution on a high-dimensional hemisphere and targeting a data distribution on $\mathbf{z} = 0$, analogous to electric field lines. Beyond physics-inspired designs[62], Lipman et al.[63] introduced conditional flow matching, a generalized framework for learning optimal transport paths between distributions. The Schrödinger bridge (SB)[18, 64, 65, 51], as an entropy-regularized optimal transport on path space, enables more efficient sampling than Langevin dynamics. Doucet et al.[20, 28] solved the SB via Iterative Proportional and Markovian Fitting, and showed that it can recover classical diffusion behaviors[25].

## 2.3 The connection of non-physics-inspired generative models to statistical physics

**Flow-based models** [33–35] construct invertible, differentiable maps from simple latent variables to complex data, enabling exact likelihoods and efficient sampling via the change-of-variables formula: $P(x) = P(z) \left| \det \left( \partial f^{-1}/\partial x \right) \right|$. Though not explicitly physics-inspired, flow models align with non-equilibrium theory. Their governing ODEs reduce to the continuity equation (see Appendix B):

$$\frac{\partial P(x,t)}{\partial t} = -\frac{\partial v P(x,t)}{\partial x},\tag{6}$$

where $v$ denotes system velocity. If $v = f - \partial \log P(x,t)/\partial x$, this becomes the Fokker–Planck equation with drift $f = v + \partial \log P(x,t)/\partial x$. Hence, flow models recover either Hamiltonian or dissipative dynamics, bridging generative modeling with non-equilibrium statistical physics.

**Variational Autoencoders** (VAEs) constitute a powerful class of generative models that leverage latent variables to represent high-dimensional data distributions [13–16, 66]. By introducing latent variables $z$ with a prior distribution $P(z)$, VAEs define a probabilistic generative process $P_\theta(x|z)$ parameterized by neural networks. Due to the intractability of the marginal likelihood $P_\theta(x)$, VAEs approximate the posterior with a variational distribution $Q_\phi(z|x)$, and optimize the evidence lower bound (ELBO) [13, 67]:

$$\log P_\theta(x) \geq \mathbb{E}_{Q_\phi(z|x)}[\log P_\theta(x|z)] - D_{KL}(Q_\phi(z|x)||P(z)).\tag{7}$$

Next we will show that the ELBO takes the same mathematical form of variational free energy $F$ in equilibrium systems [68], which satisfies $F = -kT \ln Z$, where $Z = \int \exp[-E(z)/kT]\mathrm{d}z$ is the partition function of Boltzmann distribution in Equation 1. However, the real Boltzmann distribution is hard to obtain; hence, we approximate it by a variational distribution $Q(z)$. Then, we can define the variational free energy of the equilibrium system

$$F(Q) = \mathbb{E}_Q(z)[E(z)] - kTH(Q) \geq F,\tag{8}$$

where $H(Q)$ is the variational entropy of the system. Compared with Equation 7 with 8, the ELBO parallels variational free energy, with its likelihood and KL terms corresponding to expected energy and entropy, thereby linking VAEs to both equilibrium and non-equilibrium statistical mechanics [16].

**Autoregressive models.** Unlike flow-based model and VAE, autoregressive models have little relation with physics [36, 69, 37, 38, 70]. Their main idea is to decompose high-dimensional data distributions into products of conditional probabilities using the chain rule:

$$P(\mathbf{x}) = \prod_{i=1}^{n} P(x_i|x_{<i}).\tag{9}$$

Autoregressive models have achieved great success in text and image generation [36, 69, 37, 38, 70–73]. Autoregressive models have shown promise in video generation [74], but often produce artifacts that violate physical laws due to their lack of explicit temporal dynamics or causal structure. Unlike physics-informed models (e.g., Langevin dynamics or neural ODEs/SDEs), they capture statistical correlations without modeling system evolution, making them insufficient for representing real-world dynamical processes without additional structural constraints.

**Generative Adversarial Networks** (GANs)[39] synthesize high-quality samples by training a generator and discriminator adversarially, bypassing explicit likelihood modeling. While successful in image generation[75–78], GANs lack mechanisms to model temporal evolution or causality. Since samples are independently drawn from latent space, they fail to capture time-dependent consistency or state transitions, making them unsuitable for dynamic tasks like video synthesis or trajectory prediction without architectural or hybrid modifications.

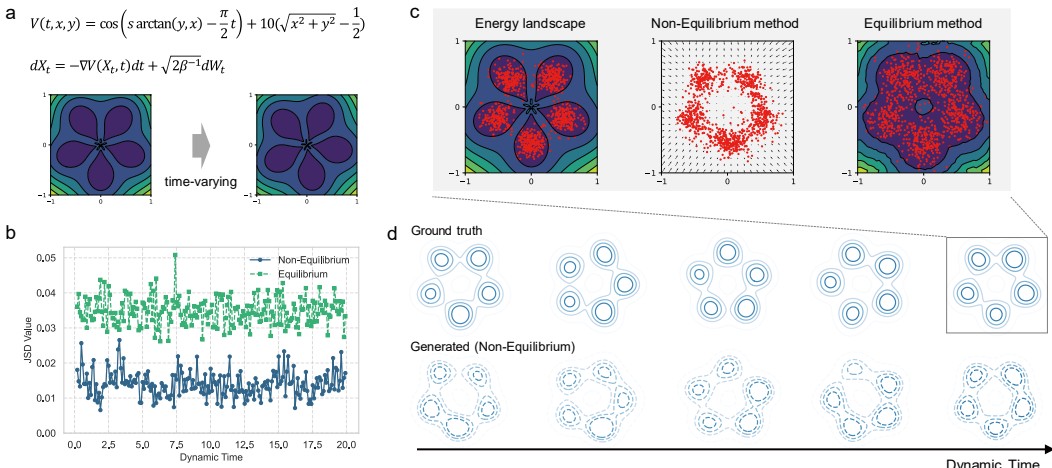

Figure 1: Equilibrium vs. non-equilibrium generation on a time-varying potential system. (a) Potential & dynamics equations. (b) Generation error comparison. (c) True energy landscape (left), non-equilibrium gradient field (mid), equilibrium energy field (right). (d) Generated distribution by non-equilibrium method.

## 3  Non-Equilibrium Generative Modeling Better Captures Dynamics in Evolving Systems

In this section, we demonstrate two viewpoints through experiments: (1) non-equilibrium dynamics are better suited for modeling systems without well-defined energy landscapes, such as those that remain far from equilibrium or exhibit only transient steady states; (2) non-equilibrium generative models more effectively capture dynamic behaviors and temporal evolution, aligning with the intrinsic time-asymmetry of many real-world systems.

To directly compare the differences in generative models guided by the principles of equilibrium and non-equilibrium processes for generative modeling of complex systems, we modify and conduct tests on a 2-dimensional Printz potential system (Figure 1a) in relevant literature [79, 80]. The particle's motion follows overdamped Langevin dynamics, with its drag term described by the potential energy landscape. The governing equation of potential function involves a time-dependent term, which leads to a time-varying energy landscape (rotating). As the potential energy changes over time, the particle's motion is continuously perturbed, causing the system to remain in non-equilibrium transitions between different states. Our generative modeling objective is to learn the time-varying energy field $V(t, x, y)$ and subsequently generate reliable particle distributions $p(x, t)$. Inspired by the distinctions between equilibrium and non-equilibrium processes, we adopt two approaches for modeling the energy field: i) Equilibrium method, and ii) Non-equilibrium method. We simulate particle trajectories under 2,000 initial conditions for training and testing, using the Jensen-Shannon Divergence (JSD) of the generated particle distributions as the error metric. Detailed methods and settings can be found in Appendix D.

In generated time-varying particle distributions, the JSD of the non-equilibrium method consistently remains lower than that of the equilibrium method (Figure 1b), highlighting its strong generative modeling potential for complex systems. Figure 1c visualizes the true energy landscape at a dynamic moment and the predictions from both methods. The non-equilibrium method uses a gradient field to reflect the energy function's influence on particles, sampling the true distribution through a constructed non-equilibrium process (denoising Langevin dynamics). In contrast, the equilibrium method simply predicts time-varying energy values and performs Boltzmann distribution sampling. Evolving open systems often exhibit time-varying energy landscapes, rarely achieving or only briefly maintaining equilibrium. Non-equilibrium methods do not rely on the system reaching equilibrium but dynamically track the evolution of energy gradients. By aligning the generative process with the inherent non-equilibrium characteristics of the data, non-equilibrium methods accurately model the static distribution at specific moments and successfully capture the system's temporal evolution (Figure 1d). Code and data for reproduction are open source[1].

---

[1]Anonymous repository.

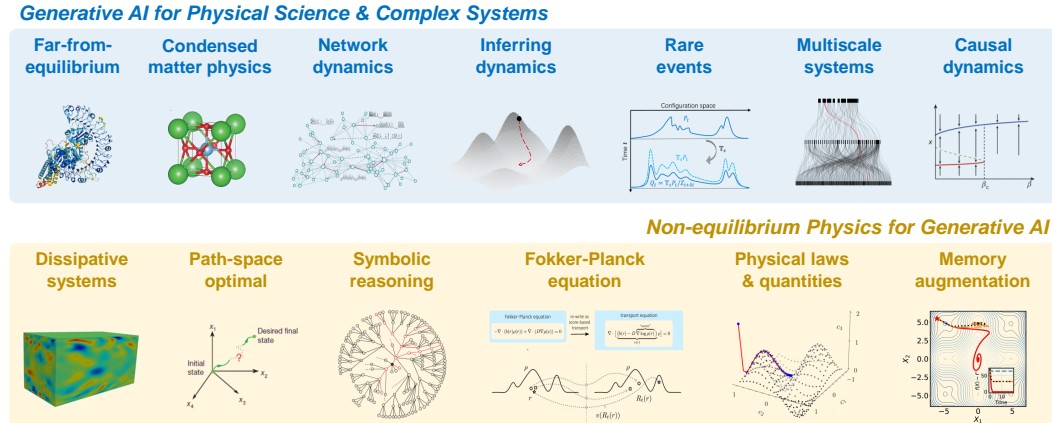

Figure 2: Outlook or future directions of both non-equilibrium physics and generative AI.

## 4    Outlook or future directions

In this section, we discuss opportunities and future directions for utilizing non-equilibrium statistical mechanics in generative AIs, mainly regarding two open questions: (i) In what fields will generative algorithms inspired by non-equilibrium physics have potential great applications? (ii) Can we utilize the wealth of non-equilibrium knowledge to develop more advanced generative models?

### 4.1    Potential future application to physical science and complex systems

**Modeling far-from-equilibrium dynamics via non-equilibrium generative processes.** A fundamental challenge in scientific modeling is capturing dynamics far from equilibrium [81–87], where systems exhibit transient, nonstationary, and strongly nonlinear behavior far from steady-state regimes. Most current generative frameworks inspired by non-equilibrium physics, such as Langevin dynamics and score-based diffusion models, implicitly assume convergence to a stationary distribution. However, many real-world systems, including climate extremes, biological differentiation, and socio-technical transitions, evolve in regimes where no steady state exists. Modeling such systems requires generative processes that track non-stationary distributions over time, adapt to evolving drift fields, and encode entropy production or phase transitions explicitly. Future work may extend generative stochastic dynamics to incorporate time-dependent control parameters or auxiliary slow-fast variables, enabling generative inference over non-equilibrated trajectories. Moreover, integrating Fokker–Planck solvers with neural sequence models could offer tractable and expressive tools. This direction opens a pathway toward generative modeling not merely of statistical distributions, but of the full unfolding of irreversible, history-dependent physical processes.

**Non-equilibrium generative models for condensed matter physics.** Diffusion generative models are proving effective in condensed matter physics for modeling high-dimensional, non-equilibrium systems. They enable unsupervised synthesis of microstructures—from crystalline and amorphous phases to defect networks like dislocations—by progressively denoising random noise into physically plausible states. In free-energy landscape sampling, diffusion processes efficiently explore multimodal surfaces, identifying equilibrium basins and transition states [79]. By conditioning on parameters such as temperature or stress, they simulate phase transitions and interface evolution, including nucleation and grain growth [86, 88–90]. In multiscale material design [91, 92], these models bridge coarse and atomistic descriptions, optimizing alloy compositions. They enhance Monte Carlo and molecular dynamics via learned proposal distributions in Metropolis–Hastings schemes, accelerating sampling for long-timescale processes like glass relaxation. Diffusion models also aid inverse problems and data fusion, reconstructing 3D atomic structures [93, 94] and revealing underlying forces. With improved computational resources, they are poised to advance materials discovery [95, 96] and non-equilibrium phase transition modeling.

**Integrating non-equilibrium generative models with complex network dynamics.** Combining non-equilibrium generative algorithms with complex network dynamics provides a powerful framework for modeling real systems [97, 98]. In this hybrid approach, a non-equilibrium generative model captures statistical laws over node and edge attributes, while network evolution is governed by

time-varying graph operators or neural dynamics (e.g., graph Neural ODEs). Coupling the two creates a feedback loop: generative samples reshape network topology, and the evolving network modulates generative dynamics. This enables realistic synthesis of temporal graphs with non-stationary structure and complex phenomena like community formation or cascading failures. Applications span epidemic forecasting [99], information diffusion, and resilient infrastructure modeling [100]. As stochastic graph flows and physics-informed graph learning advance [101], this integration will deepen our understanding of networked non-equilibrium phenomena [90].

**Inferring dynamics via non-equilibrium generative models** Non-equilibrium generative models have primarily focused on reproducing complex data distributions from dynamic systems. A promising frontier is inverting this goal: using generative processes to infer hidden dynamical mechanisms [102–104]. Models such as score-based SDEs, Fokker–Planck frameworks, and Schrödinger bridges encode interpretable quantities like drift fields, entropy production, and dissipation, linking data trajectories to physical insights. Parameterizing dynamics via physics-informed neural architectures enables principled inference of evolution laws. This supports causal discovery [105], intervention analysis [106, 107], and mechanism-aware simulation in non-stationary systems. In particular, these models may recover drivers of rare events or phase transitions where equilibrium methods fail. Developing robust, physically grounded inverse modeling tools presents both a challenge and an opportunity for integrating AI with non-equilibrium statistical physics.

**Non-equilibrium generative AIs for rare events.** Rare events in complex systems, such as extreme climate [108] or abrupt phase transitions [83, 86, 109–111], are hard to sample and predict due to their low probability and dynamic complexity. Non-equilibrium generative models can be adapted to generate trajectories biased toward rare-event regions by conditioning on boundary states, modifying drift fields, or minimizing entropy-regularized action costs. Incorporating non-equilibrium principles [54, 31, 56, 32, 112, 113, 42] further enhances physical fidelity and sampling efficiency. This enables not only targeted generation of rare transitions but also estimation of their probabilities and underlying mechanisms. Such approaches are well-suited for scientific applications where traditional Monte Carlo methods fail, offering a new generative framework for forecasting, simulating, and interpreting rare events in non-stationary environments [114, 115].

**Non-equilibrium generative AI for multi-scale systems** Multi-scale structures are ubiquitous in real-world physical systems [116], e.g., the Rayleigh–Bénard convection in the thermal fluid systems [117], the dense particle clusters surrounded by the dilute broth in the gas-solid two-phase flow [118, 119], and hierarchical folding structure of proterns [120]. Generative AIs excel at modeling multi-scale structures of real-world data through guidance loss incorporation [121], posterior reconstruction [122], or conditional modeling [123]. As a result, they provide a powerful framework for understanding and capturing the multi-scale features of complex physical phenomena. In particular, renormalization-group-inspired generative models [124] hierarchically decompose complex systems via block transformations and multi-scale latents for efficient synthesis. In quantum many-body physics, coarse latents capture collective modes while finer latents reconstruct entanglement networks [125]. Financial modeling uses scale-invariant flows to generate market regimes across time horizons. Epidemiology employs layered generators to simulate disease spread from global patterns down to local outbreaks [126]. Protein folding [127, 128] benefits from wavelet-based latent layers that first form secondary structures, then refine atomic contacts. Ecological networks [79] adopt recursive block flows to model interactions from ecosystem down to community scales.

**Modeling causal dynamics through non-equilibrium-inspired generative process** A promising future direction is to extend non-equilibrium generative models toward causal reasoning [19, 31, 56, 107, 106, 129–132], specifically, modeling interventions and counterfactuals in complex dynamical systems, which is similar to the idea of building a phase diagram in thermodynamics. Many physical and biological processes evolve under structured external influences, where interventions (e.g., perturbing a force field or blocking a biochemical reaction) induce non-trivial changes in system trajectories. While current generative models capture statistical correlations or average trajectories, they do not inherently simulate responses under hypothetical or counterfactual scenarios. Embedding causal structure into generative dynamics, by parameterizing intervention operators within stochastic differential equations, or learning structural-functional factorizations over time, could enable sampling from correlational distributions. Bridging score-based diffusion models with recent advances in causal dynamics, and incorporating thermodynamic constraints on entropy flow under intervention, may allow generative models what could have been under alternative dynamic laws. This direction opens new possibilities for scientific modeling, policy design, and decision-making in complex systems.

## 4.2 Non-equilibrium physics for more advanced generative models

**Langevin dynamics integrated non-equilibrium EBM generative model for dissipative systems.**
Unlike equilibrium EBMs that satisfy detailed balance and model data via a Boltzmann distribution
reflecting purely conservative interactions, non-equilibrium EBMs offer a principled bridge between
non-equilibrium statistical mechanics and generative AI by viewing data as states of driven, dissi-
pative systems. In this setting, the energy function encodes both conservative and non-conservative
forces, yielding a time-dependent model distribution $P_t(x) \propto e^{-E_t(x)}$. Training typically relies on
contrastive divergence or score matching, avoiding the need to compute the intractable partition func-
tion. Langevin dynamics serves as a natural tool for both sampling and learning, iteratively combining
gradient descent with noise to explore low-energy regions. Importantly, one can embed physical
priors—such as conservation laws (mass, momentum, energy) and symmetry constraints (e.g., trans-
lational, rotational)—into the architecture or dynamics [133–139], improving generalization and
interpretability. These models have been applied to learning turbulent flow under Navier–Stokes
dynamics [140] and reconstructing free-energy landscapes in complex reactions [141, 79]. Remaining
challenges include estimating high-dimensional time-varying partition functions, ensuring long-term
consistency, and reducing the computational cost of Langevin-based sampling.

**Non-equilibrium processes for path-space optimal generative models** A promising research
direction lies in modeling distributions on path space, the space of time-indexed trajectories, by
leveraging variational principles from non-equilibrium statistical mechanics. In particular, the
Schrödinger bridge problem [18–20] and entropy-regularized optimal transport [142, 143] offer a
theoretical foundation for learning the most probable evolution between prior and target distributions
under stochastic dynamics. Future generative frameworks may adopt dynamic score matching,
stochastic control, or time-dependent diffusion bridges to sample from such path-space distributions.
These methods can capture time-asymmetry, non-Markovianity, and transient structures in complex
systems. Moreover, parameterizing drift and diffusion fields in neural SDEs with physical constraints
can enable controllable and interpretable generation of trajectories. Path-space generative modeling
thus opens a new frontier in AI for science, with applications in rare-event simulation, path transition,
and dynamical inference in multiscale systems [144]. Developing solvers for such high-dimensional,
temporal generative problems remains a core challenge and opportunity.

**Bridging non-equilibrium generative models and symbolic reasoning for scientific discovery.**
Non-equilibrium generative models have demonstrated remarkable capacity to simulate complex
dynamics in high-dimensional systems. Yet, their learned representations often remain opaque,
hindering interpretability and scientific generalization. A promising direction is to integrate these
models with symbolic reasoning frameworks [145]. Specifically, non-equilibrium models can generate
rich trajectory data or latent fields that reflect underlying dynamics. These outputs can then serve as
substrates for symbolic regression that extract interpretable laws from data [146, 147]. Conversely,
symbolic priors such as conservation laws, symmetry constraints, or algebraic invariants can be
encoded into the learning objective or architecture of the generative model, guiding it toward plausible
dynamics. This synergy enables a bidirectional interface: using symbolic structures to constrain and
interpret neural models, and using generative models to ground and test symbolic hypotheses. Such
integration holds transformative potential for discovering interpretable models of non-equilibrium
phenomena in physics, biology, and beyond.

**Neural ODEs/SDEs integrated Fokker–Planck–based model for non-equilibrium systems.**
Combining Fokker–Planck–based generative modeling with Neural ODEs/SDEs yields a uni-
fied, physics-informed framework for learning and sampling complex, non-equilibrium distribu-
tions [20, 148]. In this approach, both drift and diffusion processes are parameterized by neural
networks, and sample trajectories evolve under a learned stochastic process whose probability density
obeys Fokker–Planck dynamics. The model treats continuous time evolution as a normalizing flow,
enabling exact density tracking and invertible transformations. Training proceeds by maximizing the
data likelihood or matching score fields via the adjoint method so that the learned dynamics trans-
form a simple prior distribution into the target non-equilibrium steady state. This scheme captures
both microscopic particle evolution and macroscopic density evolution, ensuring generated samples
respect physical conservation laws and transient behaviors. Applications span plasma fluctuation
synthesis [149] and simulating many-body dynamics [150]. By fusing Fokker–Planck dynamics
with continuous-time neural solvers, this hybrid strategy offers a principled path for high-fidelity,
data-driven simulation and control of driven, dissipative systems.

**Introducing non-equilibrium physical quantities and laws into generative models.** Though diffusion models can be described without non-equilibrium dynamics, non-equilibrium thermodynamics yields crucial insights. Physical quantities, such as free energy, score functions, mutual information [151], and entropy production rate, map exactly onto diffusion metrics, enabling effective solutions for real-world tasks such as high-dimensional density estimation [152]. These mappings facilitate tackling non-equilibrium challenges by calibrating diffusion parameters and improving sampling efficiency. Moreover, established non-equilibrium theories guide the design of noise schedules and denoising functions: for example, Ikeda et al. [153] quantified a trade-off between entropy production and generation error—measured via Wasserstein distance—informing the development of efficient diffusion mechanisms. Moreover, it is potentially viable to optimize generative sampling by incorporating dissipated work and non-equilibrium work theorems into model training. Batch-estimate average dissipated work or Jarzynski objectives [54], dynamically adjust sampling drift during training, and apply Crooks path-probability regularization [56, 154] to accelerate convergence and improve efficiency.

**Non-equilibrium physics for memory augmentation.** Memory augmentation in generative models leverages non-equilibrium physics by embedding history-dependent feedback into latent dynamics. One approach uses delay kernels that convolve past latent states with learnable weights, capturing viscoelastic and hysteretic behavior. Alternatively, fractional derivative operators [155] introduce power-law memory across long timescales, modeling persistent correlations in fluctuation–dissipation processes. Autoregressive latent structures incorporate time lags, enabling the model to reference past states and approximate non-Markovian drift [156–158]. These mechanisms equip generative models with long-range memory, enabling accurate simulation of slow relaxation and driven steady states while preserving invertibility.

## 5 Alternative Views

While we advocate for integrating non-equilibrium statistical mechanics into generative modeling, several alternative perspectives merit discussion:

**Empirical success of non-physics-inspired and equilibrium-based models** might suggest that they are sufficient for scientific applications. This raises the question: why introduce non-equilibrium physics at all? We acknowledge these models' success, but emphasize that the target system resides near a stationary regime or violates physical constraints. For dynamics systems, non-equilibrium models offer a principled route to temporal evolution; hence, it is more faithful to the temporal structure and entropy-producing nature of dynamic systems.

**Computational complexity concerns.** Non-equilibrium-inspired models could lead to significant training and sampling costs. Critics may view this as impractical for large-scale deployment. However, recent progress in neural solvers, adaptive sampling, and variational approximations has significantly reduced such burdens. Furthermore, the gains in modeling fidelity, especially for systems far from equilibrium, justify the additional complexity in many scientific contexts.

**Questioning physical interpretability.** Some may argue that non-equilibrium-based AI methods do not guarantee interpretability in learned representations. While interpretability does not automatically emerge, it would be beneficial from the fertile ground of non-equilibrium knowledge prior. In this way, non-equilibrium models provide scaffolds for extracting mechanistic insight, not just fitting data.

In summary, while equilibrium models and neural architectures without physical grounding have proven successful, they often fall short in representing the time-dependently non-equilibrium behaviors observed in real-world dynamical systems. Non-equilibrium generative models offer a complementary and increasingly necessary paradigm.

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

# A Representative cases of generative models driven by non-equilibrium stochastic dynamics

Random variables $\mathbf{x_t}$ that evolve according to a stochastic process can be modeled as an Ito process given by

$$d\mathbf{x_t} = \mu(\mathbf{x_t}, t)dt + \sigma(\mathbf{x_t}, t)dW_t, \tag{10}$$

where $\mu$ represents the drift term, characterizing the deterministic interactions or correlations within the system or data, and $\sigma$ describes the diffusion term, capturing the inherent stochastic fluctuations. Here, $W_t$ denotes a standard Wiener process. The corresponding partial differential equation, known as the Fokker–Planck equation, can be derived from Equation (10) to describe the evolution of the probability density associated with $\mathbf{x_t}$ (see Appendix A for the detailed derivation).

$$\frac{\partial P(\mathbf{x}, t)}{\partial(t)} = -\frac{\partial}{\partial \mathbf{x}}[\mu(\mathbf{x}, t)P(\mathbf{x}, t)] + \frac{\partial^2}{\partial \mathbf{x}^2}[D(\mathbf{x}, t)P(\mathbf{x}, t)], \tag{11}$$

where probability distribution $P(\mathbf{x}, t)$ is the target of the generative model, and the diffusion coefficient $D(\mathbf{x}, t) = \frac{\sigma(\mathbf{x_t}, t)^2}{2}$. It is noteworthy that the purpose of the generative model is to generate targeted distributions $P(\mathbf{x}, t)$ from real data samples $P_{real}(\mathbf{x})$, where $P_{real}(\mathbf{x}) = P(\mathbf{x}, t = 0) = P(\mathbf{x})\delta(t)$ corresponds to the initial condition of the stochastic process.

**Example 1: Diffusion Generative Model.**

For the learning process, Diffusion model chooses the noise levels $\{\sigma_t\}_{t=1}^T$ as the diffusion coefficients of the discrete Markov process. When $T \to \infty$, it converges to a pure continuous diffusion process where the diffusion coefficient depends on time $t$,

$$d\mathbf{x_t} = \sigma(\mathbf{x_t}, \mathbf{t})dW_t. \tag{12}$$

The corresponding Fokker-Planck equation is

$$\frac{\partial P(\mathbf{x}, t)}{\partial(t)} = \frac{\partial^2}{\partial \mathbf{x}^2}[D(t)P(\mathbf{x}, t)], \tag{13}$$

where $D = \frac{\sigma(\mathbf{x_t}, \mathbf{t})^2}{2}$ and initial condition $P(\mathbf{x}, t = 0) = P(\mathbf{x})_{real}$. Diffusion generative model usually let the noise levels $\{\sigma_t\}_{t=1}^T$ as a geometric positive sequence, satisfying $\frac{\sigma_t}{\sigma_{t+1}} = C_0 > 1$, where $C_0$ is a constant. If this ratio $C_0 \sim 1$, then the prior distribution $P(\mathbf{x}, t \to \infty) = P(\mathbf{x}, T)$ can be analytically found to follow a Gaussian distribution asymptotically. For the more general cases, $P(\mathbf{x}, t)$ can be rigorously solved by Green's function method with $G(\mathbf{x}, t|\mathbf{x}, t'), (t \geq t')$, corresponding to the transition kernel in the learning process. The nature of stochastic dynamics guarantee that $P(\mathbf{x}, T)$ is uncorrelated with $P(\mathbf{x}, t = 0) = P(\mathbf{x})_{real}$, which has been proved in previous work [62].

For the process of generating samples, it satisfies a reverse process of Equations (12) and (13)

$$d\mathbf{x_t} = -D(t)\frac{\partial \log P(\mathbf{x}, t)}{\partial \mathbf{x}}dt + \sigma(\mathbf{x_t}, \mathbf{t})d\tilde{W}_t. \tag{14}$$

Note that $s(\mathbf{x}) := \frac{\partial \log P(\mathbf{x})}{\partial \mathbf{x}}$ is the score function in [25, 23], and $\tilde{W}_t$ is the time-reversal Wiener Process. Based on Equation (14), we can obtain the generating samples from the distribution following the below Fokker-Planck equation

$$\frac{\partial P(\mathbf{x}, t)}{D(t)\partial(t)} = \frac{\partial}{\partial \mathbf{x}}[s(\mathbf{x})P(\mathbf{x}, t)] + \frac{\partial^2}{\partial \mathbf{x}^2}[P(\mathbf{x}, t)]. \tag{15}$$

**Example 2: Poisson flow generative model (PFGM).**

As demonstrated in [27], the Poisson Flow Generative Model (PFGM) represents a specific instance of the stochastic Poisson flow process. The gradient flow ODE of PFGM can be seen as a non-diffusive Ito process characterized by a Poisson-field-driven force. In the more general case, the corresponding SDE for the PFGM generation process describes the stochastic dynamics of particle flow under an electric field:

$$d\mathbf{x_t} = -\frac{\partial \tilde{\psi}(\mathbf{x})}{\partial \mathbf{x}}dt + \sigma(\mathbf{x_t}, t)d\tilde{W}_t, \tag{16}$$

and the corresponding Fokker-Planck equation

$$\frac{\partial P(\mathbf{x}, t)}{\partial(t)} = -\frac{\partial}{\partial \mathbf{x}}[E(\mathbf{x})P(\mathbf{x}, t)] + \frac{\partial^2}{\partial \mathbf{x}^2}[D(\mathbf{x}, t)P(\mathbf{x}, t)], \tag{17}$$

where $E(\mathbf{x}) = -\frac{\partial \tilde{\psi}(\mathbf{x})}{\partial \mathbf{x}}$ represents the electric field derived from the Poisson equation. By comparing Equations (16)–(17) with earlier formulations such as (14)–(15), it becomes clear that the electric field $E(\mathbf{x})$ serves as the counterpart to the score function $s(\mathbf{x})$ in the earlier example. This highlights a profound physical interpretation of the score-based method, where the score function $s(\mathbf{x})$ corresponds mathematically to the gradient of the potential field—essentially, the "free energy." This equivalence confirms that the generative process is consistent with the well-established Action Principle in physics. For the specific case of PFGM, the diffusion term $\sigma$ approaches zero, allowing it to utilize an ODE-based framework.

PFGM's training process involves perturbing real data in a manner that, while consistent with a stochastic dynamic process, is not strictly limited to a diffusion process as described by the Ito formulation in Equation (10). Unlike traditional approaches that rely on Gaussian noise, PFGM allows for noise distributions beyond Gaussian. Nevertheless, the noise scales chosen in PFGM remain closely aligned with those in *example 1*, which emphasizes the importance of points near the real data in the overall generative process. As a result, if PFGM were to sample noise from a Gaussian distribution, its forward process would recover the form given by Equation (12).

**Example 3: Diffussion Schödinger Bridge Model (DSBM).**

In comparison to *examples 1 & 2*, DSBM provides a more general generative framework that aligns with Equations (10)–(11). DSBM is rooted in the classic non-equilibrium statistical physics problem known as the Schrödinger Bridge Problem. This foundational problem seeks to characterize the non-equilibrium stochastic transportation process that transitions from an initial distribution $P(x, t = 0)$ to a target distribution $P(x, t = T)$ at time $T$. The underlying dynamics can be described by the following SDE:

$$d\mathbf{x_t} = F(\mathbf{x_t}, t)dt + \sigma(\mathbf{x_t}, t)dW_t, \tag{18}$$

where the drifting field $F(\mathbf{x_t}, t) = f(\mathbf{x_t}, t) + \sigma(\mathbf{x_t}, \mathbf{t})^2 \frac{\partial \psi(\mathbf{x})}{\partial \mathbf{x}}$. The corresponding Fokker-Planck equation satisfies

$$\frac{\partial P(\mathbf{x}, t)}{\partial(t)} = -\frac{\partial}{\partial \mathbf{x}}[F(\mathbf{x}, t)P(\mathbf{x}, t)] + \frac{\partial^2}{\partial \mathbf{x}^2}[D(\mathbf{x}, t)P(\mathbf{x}, t)]. \tag{19}$$

The time-reversal process of Equations (18) and (19) satisfies

$$d\mathbf{x_t} = \tilde{F}(\mathbf{x_t}, t)dt + \sigma(\mathbf{x_t}, t)d\tilde{W}_t, \tag{20}$$

and

$$\frac{\partial P(\mathbf{x}, t)}{\partial(t)} = -\frac{\partial}{\partial \mathbf{x}}[\tilde{F}(\mathbf{x}, t)P(\mathbf{x}, t)] + \frac{\partial^2}{\partial \mathbf{x}^2}[D(\mathbf{x}, t)P(\mathbf{x}, t)], \tag{21}$$

where $\tilde{F}(\mathbf{x}, t) = f(\mathbf{x_t}, t) - \sigma(\mathbf{x_t}, \mathbf{t})^2 \frac{\partial \tilde{\psi}(\mathbf{x})}{\partial \mathbf{x}}$. Moreover, given the ensemble distribution $P(x, t = T)$ at time $T$ and the initial condition $P(x, t = 0)$, Equations (18-21) should simultaneously satisfy $\psi(\mathbf{x_t}, t = 0)\tilde{\psi}(\mathbf{x_t}, t = 0) = P(\mathbf{x_t}, t = 0) = P_{real}(x)$ and $\psi(\mathbf{x_t}, t = T)\tilde{\psi}(\mathbf{x_t}, t = T) = P(\mathbf{x_t}, t = T)$. DSBM reframes the task of generating the ensemble distribution as one of finding an optimal energy field, represented by $\psi$ and $\tilde{\psi}$. In machine learning applications, obtaining a closed-form analytical solution for $\psi$ is infeasible. As a result, the problem of identifying $\psi$ and $\tilde{\psi}$ is treated as a reparameterization challenge. Notably, both diffusion models and PFGM can be seen as specific instances of DSBM: the diffusion model corresponds to a constant potential $\psi = c$, while PFGM is characterized by $\tilde{\psi} = E(\mathbf{x})$ and a non-zero diffusion term $\sigma \sim 0$.

# B   Generating samples by the Fokker-Planck equation.

The Fokker-Planck equation (Equation (11)) describes the distribution sample $P(\mathbf{x}, t)$ generating process, whicn can be derived from the Eq. (10). Starting with the Ito's lemma with an arbitrary integrable function $f(x)$, we obtain

$$df(\mathbf{x}_t) = \left\{ f'(\mathbf{x_t})\mu(\mathbf{x_t}, t) + \frac{1}{2}f''(\mathbf{x_t})\sigma(\mathbf{x_t}, t)^2 \right\} dt + f'(\mathbf{x_t})\sigma(\mathbf{x_t}, t)dW_t. \tag{22}$$

940 The above equation leads to

$$f(\mathbf{x}_t) = f(\mathbf{x}_t; T = 0) + \int_0^t \left\{ f'(\mathbf{x_t})\mu(\mathbf{x_t},t) + \frac{1}{2}f''(\mathbf{x_t})\sigma(\mathbf{x_t},t)^2 \right\} dt + \int_0^T f'(\mathbf{x_t})\sigma(\mathbf{x_t},t)dW_t.$$
(23)

941 Next we define the probability density function $P(\mathbf{x_t}|t)$. Incorporating $P(\mathbf{x_t}|t)$ with Eq. (23) leads
942 to

$$\int_a^b P(\mathbf{x_t}|t)f(\mathbf{x_t})d\mathbf{x_t} = \int_a^b d\mathbf{x_t} P(\mathbf{x_t}|t) \int_0^T \left\{ f'(\mathbf{x_t})\mu(\mathbf{x_t},t) + \frac{1}{2}f''(\mathbf{x_t})\sigma(\mathbf{x_t},t)^2 \right\} dt$$
$$+ \int_a^b d\mathbf{x_t} P(\mathbf{x_t}|t) \int_0^T f'(\mathbf{x_t})\sigma(\mathbf{x_t},t)dW_t + f_0(\mathbf{x_t}).$$
(24)

943 Given the expectation of the randomness term of Ito process equal to zero
944 $\int_a^b \int_0^T P(\mathbf{x_t}|t)f'(\mathbf{x_t})\sigma(\mathbf{x_t})dW_t d\mathbf{x_t} = 0$

$$\int_a^b P(\mathbf{x_t}|t)f(\mathbf{x_t})d\mathbf{x_t} = \int_a^b d\mathbf{x_t} P(\mathbf{x_t}|t) \int_0^T \left\{ f'(\mathbf{x_t})\mu(\mathbf{x_t},t) + \frac{1}{2}f''(\mathbf{x_t})\sigma(\mathbf{x_t})^2 \right\} dt + f_0(\mathbf{x_t})$$
(25)

945 Then, we take time derivative on the both side of Eq.(25)

$$\frac{\partial}{\partial t} \int_a^b P(\mathbf{x_t}|t)f(\mathbf{x_t})d\mathbf{x_t} = \int_a^b d\mathbf{x_t} P(\mathbf{x_t}|t) \left[ f'(\mathbf{x_t})\mu(\mathbf{x_t},t) + \frac{1}{2}f''(\mathbf{x_t})\sigma(\mathbf{x_t})^2 \right].$$

946 By using the partial integral method, we obtain

$$\frac{\partial}{\partial t} \int_a^b P(\mathbf{x_t}|t)f(\mathbf{x_t})d\mathbf{x_t} = P(\mathbf{x_t}|t)\mu(\mathbf{x_t},t)\Big|_a^b - \int_a^b d\mathbf{x_t} \frac{\partial[P(\mathbf{x_t}|t)\mu(\mathbf{x_t},t)]}{\partial \mathbf{x_t}} f(\mathbf{x_t})$$
$$+ \frac{1}{2}P(\mathbf{x_t}|t)f'(\mathbf{x_t})\sigma(\mathbf{x_t},t)^2\Big|_a^b - \frac{1}{2} \int_a^b d\mathbf{x_t} \frac{\partial(P(\mathbf{x_t}|t)\sigma(\mathbf{x_t})^2)}{\partial \mathbf{x_t}} f'(\mathbf{x_t}).$$

947 Notice that $P(\mathbf{x_t}|t)\mu(\mathbf{x_t},t)\Big|_a^b = 0$ and $\frac{1}{2}P(\mathbf{x_t}|t)f'(\mathbf{x_t})\sigma(\mathbf{x_t},t)^2\Big|_a^b = 0$. Therefore, we finally obtain

$$\frac{\partial}{\partial t} \int_a^b P(\mathbf{x_t}|t)f(\mathbf{x_t})d\mathbf{x_t} = - \int_a^b d\mathbf{x_t} \frac{\partial[P(\mathbf{x_t}|t)\mu(\mathbf{x_t},t)]}{\partial \mathbf{x_t}} f(\mathbf{x_t}) + \int_a^b d\mathbf{x_t} \frac{\partial^2(P(\mathbf{x_t}|t)\sigma(\mathbf{x_t})^2)}{\partial \mathbf{x_t}^2} f(\mathbf{x_t}).$$
(26)

948 The above equation is the integral formulation of Fokker-Planck equation. For the three representative
949 examples, their Fokker-Planck equation exhibits different drifting and diffusion behaviors, we
summarize them in the following table

Table 2: Summary of the drifting and diffusion terms of generative models.

| Models | Forward Drifting | Forward Diffusion | Backward Drifting | Backward Diffusion |
|---|---|---|---|---|
| DGM | 0 | $\sigma(\mathbf{x_t},\mathbf{t})$ | $-\frac{\sigma(\mathbf{x_t},\mathbf{t})^2}{2}s(\mathbf{x})$ | $\sigma(\mathbf{x_t},\mathbf{t})$ |
| PFGM | NA | NA | $E(\mathbf{x})$ | $\sim 0$ |
| SBM | $\sim \sigma(\mathbf{x_t},\mathbf{t})^2\frac{\partial\psi(\mathbf{x})}{\partial\mathbf{x}}$ | $\sigma(\mathbf{x_t},\mathbf{t})$ | $\sim -\sigma(\mathbf{x_t},\mathbf{t})^2\frac{\partial\psi(\mathbf{x})}{\partial\mathbf{x}}$ | $\sigma(\mathbf{x_t},\mathbf{t})$ |

950

# C  Asymptotical independency of the initial state for the Brownian motion

The Brownian motion is a typical stationary independent incremental stochastic process, where $\mathbf{x_{t_0}}, \mathbf{x_{t_1}} - \mathbf{x_{t_0}}, \ldots, \mathbf{x_{t_n}} - \mathbf{x_{t_{n-1}}}$ are independent and identical distributed for $t \in [0, \infty)$. This indicates that the (drifting) Brownian motion is characterized by the Green's function $G(\mathbf{x}, t | \mathbf{x_0}, 0)$, which corresponds to the transition probability of generative models. Note that only non-drifting process satisfy $G(\mathbf{x}, t | \mathbf{x_0}, 0) = \frac{1}{\sqrt{2\pi\sigma^2 t}} e^{\frac{-(\mathbf{x} - \mathbf{x_0})^2}{2\sigma^2 t}}$, which is the special case for drifting Brownian motion. For the other drifting process, $\frac{1}{\sqrt{2\pi\sigma^2 t}} e^{\frac{-(\mathbf{x} - \mathbf{x_0})^2}{2\sigma^2 t}}$ is the first order term of their Green's function. Therefore, we have

$$G(\mathbf{x}, t | \mathbf{x_0}, 0) \sim \frac{1}{\sqrt{2\pi\sigma^2 t}} e^{\frac{-(\mathbf{x} - \mathbf{x_0})^2}{2\sigma^2 t}} \tag{27}$$

Given the Fokker-Planck equation, we can always write down its analytical solution in an integral form by using the Greens function, following

$$P(\mathbf{x}, t) = \int \mathrm{d}\mathbf{x} P(\mathbf{x_0}, 0) G(\mathbf{x}, t | \mathbf{x_0}, 0). \tag{28}$$

Considering a large $t = t_\infty \to \infty$, (27) indicates that $G \sim \frac{1}{\sqrt{2\pi\sigma^2 t}}$, which means the Green's function are governs by large time $t$. This further indicates

$$P(\mathbf{x}, t \to \infty) = \int \mathrm{d}\mathbf{x} P(\mathbf{x_0}, 0) e^{\frac{-(\mathbf{x} - \mathbf{x_0})^2}{2\sigma^2 t_\infty}} = \int \mathrm{d}\mathbf{x} P(\mathbf{x_0'}, 0) e^{\frac{-(\mathbf{x} - \mathbf{x_0'})^2}{2\sigma^2 t_\infty}}, \tag{29}$$

where $P(\mathbf{x_0}, 0)$ and $P(\mathbf{x_0'}, 0)$ are different initial distributions. (29) means the stationary state $P(\mathbf{x}, t \to \infty)$ of drifting Brownian motion are independent of their initial states because the Greens function's asymptotic behavior for is dominated by large $t$. This aligns with the requirement of generative model: the prior distribution $P(\mathbf{x}, t \to \infty)$ is independent of the real data $P_{real}(\mathbf{x})$.

# D    Experiment on a time-varying potential system

To directly compare the differences in generative models guided by the principles of equilibrium and non-equilibrium processes for generative modeling of complex systems, we modify and conduct tests on a 2-dimensional dynamic system in relevant literature [79, 80].

This system is a Printz potential well within a 2-dimensional bounded region, governed by the equation

$$V(t, x, y) = \cos\left(s \arctan(y, x) - \frac{\pi}{2}t\right) + 10\left(\sqrt{x^2 + y^2} - \frac{1}{2}\right), \tag{30}$$

$$dX_t = -\nabla V(X_t, t)dt + \sqrt{2\beta^{-1}}dW_t, \tag{31}$$

where $s = 5$ defines the number of potential wells around the origin and $\beta = 10$ controls the noise intensity. The time-dependent term in the potential function leads to a time-varying energy landscape with a period of $\pi/2s$. This simulates the system's response to periodic external disturbances, serving as a simplified representation of real-world scenarios. The particle's motion follows overdamped Langevin dynamics, with its drag term described by the potential energy landscape. As the potential energy changes over time, the particle's equilibrium state is continuously perturbed, causing the system to remain in non-equilibrium transitions between different states.

Our generative modeling objective is to learn the time-varying energy field $V(t, x, y)$ and subsequently generate reliable particle distributions $p(x, t)$. Inspired by the distinctions between equilibrium and non-equilibrium processes, we adopt two approaches for modeling the energy field:

- **Equilibrium method** assumes that the particle distribution at each time instance adheres to a Boltzmann distribution defined by the energy [141]. We construct an approximately static energy field by statistically analyzing sample frequencies. The static energy field at each time instance is then used to supervise the training of a parameterized time-varying neural field $V_\theta(t, x, y)$. This process distills the continuously evolving energy field into a parameterized model for continuous sampling during testing.

- **Non-equilibrium method** focuses on learning the particle distribution as a function of time, $p(x|t)$. We employ a conditional diffusion model [25], incorporating the dynamic time $t$ as an additional conditional input to the parameterized score function $f_\theta(x_n, n)$. This enables the model to learn the system's time-varying energy gradient directly from continuous dynamic trajectories.

We uniformly sample 2,000 initial positions within the region $[-1, 1]^2$ and simulate a total duration of $T = 20.0$ with a unit time step of $dt = 0.02$. Half of these 2,000 trajectories are used for training, and the other half for testing. We use Jensen-Shannon divergence (JSD) as the error metric for generated particle trajectories.

# E    Generative models in various physics systems

**Molecular System.** Molecules are commonly represented as graphs, with nodes representing atoms and edges representing chemical bonds. Conformation prediction is a task that determines the three-dimensional (3D) coordinates of all atoms in a given molecule, which offers a more intrinsic representation of molecules. Traditionally, this task relies on molecular dynamics simulations, which sequentially update the coordinates of atoms based on the forces acting on each atom. This process can be linked to non-equilibrium thermodynamics by considering the score function as pseudo-forces that guide atoms toward high-likelihood regions. Shi et al. [159] propose ConfGS that uses a graph isomorphism network to estimate the score function and achieves state-of-the-art (SOTA) performance. However, it does not account for long-range atomic interactions like van der Waals forces. To address this, Luo et al. [160] propose to use the graph constructed dynamically based on spatial proximity, which improves performance in predicting protein side chains and multi-molecular complexes. These methods add noise to distance matrices, which can violate the triangular inequality or lead to negative values. Therefore, Xu et al. [94] propose to add noise to atomic coordinates instead of distance matrices, significantly improving the success rate. Jing et al. [161] suggest applying the diffusion and reverse processes solely to the chemical bond torsions since it has the most significant impact on conformations, outperforming traditional cheminformatics methods for the first time.

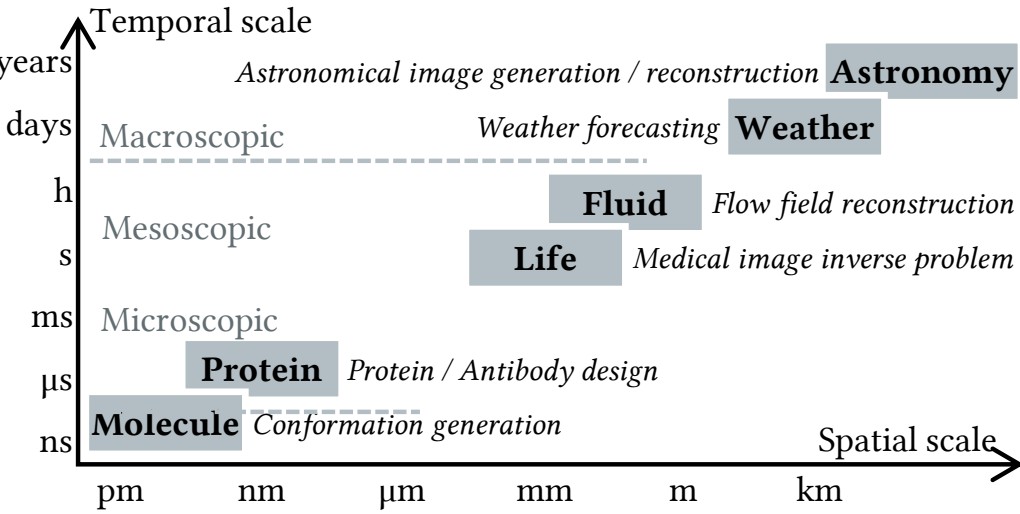

Figure 3: Physical systems invoke non-equilibrium thermodynamics and the associated problems.

**Protein System.** Proteins are biological macromolecules composed of amino acids. The *sequence* of amino acids determines its three-dimensional *structure*, which further dictates its *function*. The protein design task aims to generate protein sequences and/or structures with specific functions. Anand and Achim [162], Luo et al. [163], Watson et al. [164], Ingraham et al. [165] propose diffusion-based methods for generating protein sequences and structures, while Lee et al. [166] presents a score-based method. Anand and Achim [162] introduce a specialized diffusion scheme to generate amino acids sequences in discrete space and side chain orientations in non-Euclidean space. Luo et al. [163], Watson et al. [164] focus on generating proteins conditioned on richer prompts, such as specific antigen structures and functional descriptions. Meanwhile, Ingraham et al. [165] proposes an improved model architecture that reduces time complexity when generating proteins or protein complexes. Wu et al. [127] links the denoising diffusion process to the physics process by which proteins fold to minimize energy. Instead of working with atomic Cartesian coordinates, they perform the diffusion and reverse processes on bond angles and torsion between atoms in the protein backbone.

**Life System.** The observation of life systems heavily depends on medical imaging techniques such as magnetic resonance imaging (MRI) and computed tomography (CT), which capture data in Fourier space or sinogram space. However, obtaining full measurements often incurs significant time costs or excessive ionizing radiation exposure to patients. Therefore, it is essential to develop algorithms that can generate medical images from partial measurements, such as downsampled Fourier space data or sparse-view sinograms. Unlike deterministic supervised learning methods, diffusion model-based generative approaches offer advantages in generalization to out-of-distribution data [167] and in maintaining independence from the measurement process [168]. Jalal et al. [169], Song et al. [168], Chung and Ye [170], Chung et al. [167], Cao et al. [171] have developed score-based medical image reconstruction methods that outperform other supervised learning techniques. In contrast, Peng et al. [172], Xie and Li [173] have introduced methods based on denoising diffusion probabilistic models, which demonstrate superior accuracy due to their flexibility in controlling the noise distribution. Cao et al. [171] suggest performing denoising in the measurement space, ensuring consistency between the reconstructed image and acquired data.

**Fluid System.** Computational fluid dynamics (CFD) plays a crucial role in engineering and scientific applications. Despite the known governing equations of gas or liquid flows, traditional numerical methods face significant computational challenges, particularly in scenarios involving fine-grained simulations or high-Reynolds numbers. Diffusion models have recently gained popularity for refining coarse-grained flow field data or generating high-fidelity flow data due to their ability to capture the chaotic and stochastic nature of turbulence. Shu et al. [174] adopt a physics-informed neural network to learn the denoising process, which denoises the perturbed fluid field data to generate high-fidelity data. The trained model can reconstruct high-fidelity flow data from low-fidelity or sparsely measured data without retraining. Hu et al. [175] further demonstrates the power of diffusion models to generate high-fidelity flow fields in complex geometric scenarios, employing a U-Net neural network and incorporating obstacle geometry as prompts.

**Weather System.** Global weather forecasting is one of the most crucial problems in the weather system. Traditionally, weather forecasting relies on numerical weather prediction (NWP), which solves atmospheric dynamics models to generate deterministic future weather scenarios. However, given the inherent uncertainty in weather patterns, it is important not only to predict a single probable scenario of future weather $\mathbf{x}$ but also to assess the probability of various future outcomes $P(\mathbf{x})$ [122, 176]. Considering the non-equilibrium nature of global weather as a dynamical system, coupled with the ability of diffusion models to fit and sample from arbitrary distributions, these models present a promising solution for quantifying forecasting uncertainty. Li et al. [122] propose SEEDS, which, by conditioning on a few NWP-generated scenarios, recovers additional forecasting scenarios through a denoising process. Drawing inspiration from the similarity between weather data and video, they use a vision Transformer to model score functions. Price et al. [176] introduce GenCast, which employs a graph transformer designed for spherical meshes to generate future global weather conditions based on the current state. This approach significantly outperforms existing NWP models in both efficiency and accuracy for the first time. Their work demonstrates that diffusion-based methods can address the blurring issues commonly found in other deterministic machine learning models, underscoring the advantages of diffusion models in this context.

**Astronomical System.** Human understanding of astronomical systems is deeply reliant on astronomical imaging. These images, which project the three-dimensional universe onto two-dimensional planes, are inevitably affected by background starlight in addition to the celestial objects being observed. This interference can be likened to the introduction of noise into an image, establishing a natural connection between diffusion models and astronomical image processing. Drozdova et al. [177] and Sortino et al. [178] propose using the diffusion model for astronomical image denoising and synthesis, showing more efficient and effective performance than other methods.

