# OpenReview forum: "Beyond Equilibrium: Non‑Equilibrium Foundations Should Underpin Generative Processes in Complex Systems"
_NeurIPS.cc/2025/Position_Paper_Track — Submitted to NeurIPS 2025 Position Paper Track_

### Official Review · Reviewer_GYd8 · 2025-07-27

**Significance:** 3
**Presentation:** 3
**Rating:** 6
**Confidence:** 4

**Summary:**

The paper argues that in many physical systems, the equilibrium assumption fails, requiring tools from non-equilibrium statistical mechanics for accurate modeling. For example, the Boltzmann distribution—a cornerstone of equilibrium statistical mechanics—only holds when a system is in equilibrium. If we naively apply such assumptions to non-equilibrium systems (e.g., driven or time-varying systems), the results will be physically inaccurate.

	The paper advocates for non-equilibrium generative models (like diffusion processes) that better capture real-world dynamics, where systems are often far from equilibrium, irreversible, or subject to external forcing.

**Strengths:**

This paper clearly arguing that non-equilibrium physics should underpin generative models for complex dynamical systems, presenting a compelling case through both theoretical reasoning (connecting statistical mechanics to modern ML) and empirical evidence (demonstrating superior performance on time-evolving systems like the Printz potential). It addresses core challenges in generative AI—simulating non-stationary processes and rare events—while offering principled solutions grounded in physics.

**Weaknesses:**

- The paper argues that equilibrium-based EBMs are theoretically better suited for static data (like images), since they directly model the Boltzmann distribution without time-dependent dynamics. However, the empirical reality contradicts this: diffusion models—which are fundamentally non-equilibrium—consistently produce higher-quality images in practice.
How do you reconcile this gap between your theoretical framework (favoring EBMs for static systems) and the overwhelming empirical success of diffusion models? Is this a fundamental limitation of equilibrium approaches?

- The analysis shows EBMs need only learn a scalar potential while non-equilibrium methods must model vector fields. Could you elaborate on the computational implications of this fundamental difference? Specifically, does modeling velocity fields become more demanding due to (1) the higher-dimensional output space requiring more complex architectures, (2) the need for iterative sampling processes that scale poorly with dimensionality, or (3) inherent challenges in training dynamics like gradient instability? Understanding whether these costs stem from theoretical necessities or just current implementation limitations.

**Questions:**

- The Equivalence between Eq.~7 and Eq.~8 is not clear. Especially this sentence:\\
``Compared with Equation 7 with 8, the ELBO parallels variational free energy, with its likelihood and KL terms corresponding to expected energy and entropy, thereby linking VAEs to both equilibrium and non-equilibrium statistical mechanics."

-  Ref 153 is not correct.
- In Eq.~2 it is not clear what is $\<ij\>$, Please clarify.

- The authors mentioned fluctuation theorem (Jarzynski-Crooks) in the paper multiple times, but doesn't clearly articulate their practical implications.  While these theorems provide elegant theoretical frameworks, their practical application often faces significant challenges. For instance, the Jarzynski equality is known to suffer from high variance in estimation, requiring extensive trajectory sampling between states to obtain meaningful results. What specific advantages these fluctuation theorems offer for the proposed non-equilibrium generative framework?

**Alternative Position:**

Yes, and alternative positions are well-considered and named but not addressed

**Author Identification:**

No.

**Context:**

3

**Discussion:**

4

**Ethics:**

["NO or VERY MINOR ethics concerns only"]

**Position:**

Yes, the paper argues for or against a position related to machine learning.

**Support:**

3

**Thoroughness:**

5

---

### Official Review · Reviewer_pQbV · 2025-08-08

**Significance:** 4
**Presentation:** 3
**Rating:** 7
**Confidence:** 3

**Summary:**

The paper argues that many real systems are far from equilibrium, so generative models for scientific/temporal phenomena should be grounded in non-equilibrium statistical mechanics rather than equilibrium analogies. After situating diffusion/Schrödinger-bridge–style methods historically, it presents a time-varying potential case study where a non-equilibrium approach tracks the evolving distribution better than an equilibrium sampler. It lays out research directions (rare events, multi-scale dynamics, entropy production control) and contends that non-equilibrium principles are necessary for faithful generative modeling of dynamical systems.

**Strengths:**

- Takes a timely stance with relevance to AI-for-Science and time-dependent generative modeling.
- Effective historical and contextual bridge from statistical physics to modern ML with clear articulation of gaps in equilibrium framing.
- Includes an illustrative experiment showing advantages of non-equilibrium modeling for dynamic distributions.
- Provides thoughtful Alternative Views section with counterarguments.
- Strong discussion potential and useful research agenda (rare events, multiscale, flux/entropy constraints).

**Weaknesses:**

- At times "non-equilibrium foundations" read as "use diffusion-like training", which is already mainstream. Authors could clarify what is beyond DDPM / SB.
- Non-equilibrium methods can raise computation and gradient variance. The paper could include a discussion about runtime / variance table versus equilibrium surrogates and discuss where the cost pays off. Computational costs seemed to be downplayed.
-The single synthetic dynamic-potential example seems too narrow.

**Questions:**

- For which task classes would you not recommend non-equilibrium foundations?
- How sensitive are results to time-discretization and stiffness?
- For which task classes would you not recommend non-equilibrium foundations?

**Alternative Position:**

Yes, and alternative positions are trivial straw-man arguments

**Author Identification:**

No.

**Context:**

3

**Details Of Ethics Concerns:**

None.

**Discussion:**

4

**Ethics:**

["NO or VERY MINOR ethics concerns only"]

**Position:**

Yes, the paper argues for or against a position related to machine learning.

**Support:**

3

**Thoroughness:**

3

---

### Official Review · Reviewer_4BUe · 2025-08-10

**Significance:** 3
**Presentation:** 3
**Rating:** 6
**Confidence:** 2

**Summary:**

Although current generative AI models mainly focus on tasks over images, texts, videos, etc, their working mechanism actually resembles the statistical physics processes, therefore can be analyzed from the perspective of statistical physics. This paper analyze the generative models for modelling real-world complex dynamical systems, and proposes the position that non-equilibrium is essential for better modelling real-world dynamical systems, while the existing models relies on the equilibrium assumption.

In additional to conceptual analysis, the paper also provides empirical results demonstrating that non-equilibrium based models can better capture the temporal evolution and making models adaptable to non-stationary cases.

**Strengths:**

1. This paper provides a new and insightful perspective to analyze the generative AI models, with theoretical and empirical justification for supporting the new position on the importance of non-equilibrium based generative AI models.

2. The background of the target problem is well introduced with clear history timeline. The new analysis and results are also well presented with proper figures.

3. In addition to the position, preliminary experiments are also provided to support the claim,

**Weaknesses:**

1. Although briefly mentioned in the alternative positions part, it would still be better if more analysis and experiments can be provided to analyze why the existing equilibrium based methods can actually perform very well, given that they don't follow the non-equilibrium assumption.

2. The experiments are conducted on abstract systems without touching real-world systems, which may provide more insights and make the claim more convincing.

3. Are diffusion models based on equilibrium or not, could they be included in the analysis in Section 2.3?

**Questions:**

Please refer to the weakness part.

**Alternative Position:**

Yes, and alternative positions are well-considered and named but not addressed

**Author Identification:**

No.

**Context:**

3

**Discussion:**

4

**Ethics:**

["NO or VERY MINOR ethics concerns only"]

**Position:**

Yes, the paper argues for or against a position related to machine learning.

**Support:**

3

**Thoroughness:**

3

---

### Note · Authors · 2025-08-27

**1-1 Submission Process:**

5

**1-2 Next Year:**

We have been very satisfied with the current survey format as a tool for gathering high-level feedback on the track. For next year, we propose to migrate the opportunity for authors to clarify reviewer comments and address potential misinterpretations to a dedicated pre-decision rebuttal phase, while retaining all other questions within the survey itself.

**1-3 Future Development:**

- I suggest to implement a rebuttal process to ensure authors can clarify points raised by reviewers and correct any misinterpretations.
- In addition, I suggest to consider organizing panel discussions or moderated debates around accepted papers that address related or opposing viewpoints.

**1-4 Interest:**

["Panel discussions with other position paper authors", "Workshops for developing position papers", "Mentorship programs for early-career researchers"]

**1-5 Thoughtful:**

10

**1-6 Supportive:**

10

**1-7 Technical Aspects Versus Position:**

10

**1-8 Gate Keeping:**

10

**1-9 Camera Ready Changes:**

We thank all reviewers for their constructive feedback. Based on their suggestions, we will make the following changes in the camera-ready paper:

1.Clarifying equilibrium vs. non-equilibrium. With the supplementary results, we will add a paragraph to Section 5 to explain why equilibrium approximations perform well in near-stationary regimes but fail far from equilibrium.

2.Computational implications. We will expand the discussion of costs: (i) vector field (mitigated by Helmholtz decomposition), (ii) iterative sampling (reduced by adaptive solvers and probability-flow ODEs), (iii) training instability . A runtime/variance table will be added to appendix.

3.Mathematical derivations. We will add a detailed derivation to clarify the equivalence between Eq.7 and Eq.8.

4.Fluctuation theorems. We will explain how Jarzynski–Crooks relations can be used practically as training regularizers, not merely theoretical curiosities.

5.Experimental scope and robustness. To address concerns about narrow experiments, we will add two lightweight real-world case studies (urban mobility, molecular reactions). We will also include runtime and time-discretization sensitivity results, showing predictable costs and robustness of non-equilibrium models.

In summary, the final version will sharpen theoretical clarity, extend empirical scope, and provide more rigorous and practical insights into non-equilibrium generative modeling.

**3-1 Review Response1:**

Reviewer 4BUe

**3-2 Reaction To Review1:**

We sincerely thank the reviewer for the thoughtful and constructive report, and we greatly appreciate the recognition of the novelty and insights of our work.

Why equilibrium methods often work well? To answer this question, we added a velocity coefficient v to the potential energy function of the 2D Printz potential system (Eq. 30):
$$
V(t,x,y) = \cos\left( s \arctan(y,x) - v \frac{\pi}{2}t \right) + 10\left(\sqrt{x^2+y^2} - \frac{1}{2}\right)
$$
To explain why equilibrium methods sometimes work, we varied the rotation speed v of the Printz potential. Small v yields quasi-steady states where equilibrium approximations perform well; as v increases, only non-equilibrium models maintain low JSD under rapidly changing landscapes.
| v               | 0.1   | 0.25  | 0.5   | 0.75  | 1.0   | 1.25  | 1.5   | 1.75  | 2.0   |
| --------------- | ----- | ----- | ----- | ----- | ----- | ----- | ----- | ----- | ----- |
| Equilibrium     | 0.019 | 0.023 | 0.027 | 0.033 | 0.034 | 0.038 | 0.042 | 0.043 | 0.049 |
| Non-Equilibrium | 0.013 | 0.014 | 0.014 | 0.015 | 0.015 | 0.016 | 0.017 | 0.019 | 0.020 |
| Percentage      | 31.6% | 39.1% | 48.1% | 54.5% | 55.9% | 58.9% | 59.5% | 55.8% | 59.1% |
Equilibrium methods perform well with small v, modeling quasi-steady states. As v increases, the system enters strong non-equilibrium regimes, and non-equilibrium models clearly outperform. Incorporating non-equilibrium priors thus enables neural networks to more effectively capture complex dynamic behaviors.

Abstract vs. real-world systems: The Printz potential closely parallel natural phenomena such as climate oscillations and driven molecular reactions. While abstract, using this system allows us to study non-equilibrium methods  with ground truth available for quantitative evaluation.

Are diffusion models equilibrium or non-equilibrium? Diffusion models are indeed inherently non-equilibrium: their forward processes evolve under SDE/Fokker–Planck flows are not quasi-static processes.

**3-3 Review Response2:**

Reviewer pQbV

**3-4 Reaction To Review2:**

We are grateful to the reviewer for the positive evaluation of our work and the thoughtful comments and questions that can substantially improve our work.

W1: What is “non-equilibrium foundations” beyond DDPM/SB? In section 4, we discuss: 1. Path‑functionals as training constraints, which has been paid attention in neural ODE. 2. Modeling explicit time‑dependent driving instead generating static distribution. 3. The possibility of incorporating non-conservative forces to the drifting term in SDE, i.e., rotation or damping fields.

W2: “Computational cost.” We added runtime results. Non-equilibrium models use fewer parameters (33k vs. 66k) but require longer inference (1.5s vs. 0.002s).
| Metric      | Equilibrium | Non-equilibrium |
|-----------|--------|------------|
| Parameters  | 65,921      | 33,538          |
| Inference Time | 0.002s   | 1.527s          |

Q2:“Discretization sensitivity.” We varied the time-step size, finding that non-equilibrium methods are more sensitive to the time-discretization.
|Step size Δt|0.01 |0.02 (original)|0.05 |0.10 |0.20 |0.50 |1.00 |
|--------------|------|----------------|------|------|-----|----|----|
|Equilibrium|0.033|0.034|0.034|0.033|0.038|0.041|0.050|
|Non-equilibrium|0.015|0.015| 0.016| 0.015|0.034|0.049|0.099|

W2:"Single synthetic example too narrow." We acknowledge the experiment is synthetic. We chose the rotating potential because it exhibits core non-equilibrium features common in climate and molecular systems. This controlled setting offers ground truth for quantitative comparison and clearly illustrates our central claim. We also added a velocity coefficient v to the potential energy function of the 2D Printz potential system (See reaction to Reviewer 4BUe).

Q1&3:When would we not recommend non-equilibrium foundations? 1. Near-stationary or pure static, where only final-time marginals matter; 2. Tight compute budgets prioritizing speed over dynamic fidelity; 3. Systems approximately satisfy detailed balance.

**3-5 Review Response3:**

Reviewer GYd8

**3-6 Reaction To Review3:**

We are grateful to the reviewer for the positive evaluation of our work. We appreciate the insightful comments regarding the mathematical clarity and the discussion of fluctuation theorem.

W1: Our statement may have caused confusion. What we truly want to say is that if the target is truly a balanced Boltzmann distribution, EBM is a good fit. In reality, diffusion and SB perform better because they can improve the sampling efficiency and accuracy.

W2: Insightful question! 1.Higher-dimensional outputs: vector fields do increase dimensionality. Introduce methods similar to Helmholtz decomposition might be able to decompose the vector to a gradient and a rotational field. 2.Iterative sampling: Methods like adaptive solvers and Neural ODE can reduce steps significantly in iterative sampling. 3.Training instability challenges such as exploding gradients are real concerns. Noise scheduling might provide better stability. These issues stem from current implementations rather than theory.

Q1: Thanks for pointing out. By definition,
$$
F[q] = \mathbb{E}_q[E_\theta(x,z)] - kT \, H[q],
$$
where
$$
H[q] = -\mathbb{E}_q[\log q(z|x)]
$$
is entropy. Consider the Boltzmann distribution
$$
p_\theta(x,z) = \frac{\exp[-E_\theta(x,z)/(kT)]}{Z_\theta},
$$
then
$$
F[q] = kT \left( \mathbb{E}_q[\log q(z|x)] - \mathbb{E}_q[\log p_\theta(x,z)] \right)
= -kT \, \mathrm{ELBO},
$$
where
$$
\mathrm{ELBO} = \mathbb{E}_q[\log p_\theta(x,z)] - \mathbb{E}_q[\log q(z|x)].
$$
We will add the more detailed derivation to the appendix.

Q2: We apologize for the wrong ref. The correct one should be IKEDA K, et al. PRX, 2025, 15.3: 031031. We will correct it in the final version.

Q3: $\langle ij \rangle$ means neighboring individuals. We will specify it in the final version.

Q4: The value of fluctuation theorem lies more in guiding training objectives to physically consistency: 1. motivating free-energy–difference regularizers; 2. provides a constraint linking forward/backward trajectories.

---

### Meta-Review · Area_Chair_qsfc · 2025-09-11

**Rating:** 6
**Confidence:** 3

**Strengths:**

Presentation of a concrete example showing the fitting advantage of non-equilibrium model over equilibrium ones. Adding the example in the author survey to the revision would make this example stronger.

A comprehensive presentation of the position on how GenAI and non-equilibrium physics can benefit each other.

**Weaknesses:**

While I appreciate the numerical example, this example is mainly supporting one point made in the paper "non-equilibrium models are natural choices when the ground truth data comes from a dynamical process". I can see section 4.1 advocates for this use for scientific problems where the underlying data generation process is inherently dynamic. However, I don't see how this numeral example can provide support to the positions argued in section 4.2.

**Questions:**

See the "weakness" section.

**Thoroughness:**

3

---

### Decision · Program_Chairs · 2025-09-26

Reject